# Retrospective Study of Lung Cancer: Evolution in 16 Years in the Burgos Region

**DOI:** 10.3390/jcm13061601

**Published:** 2024-03-11

**Authors:** Gustavo Gutiérrez Herrero, Sandra Núñez-Rodríguez, Carla Collazo, Álvaro García-Bustillo, Jerónimo J. González-Bernal, Lucía Castellanos-Romero, Josefa González-Santos

**Affiliations:** 1Pulmonology Unit, Burgos University Hospital, 09006 Burgos, Spain; fggutierrez@ubu.es (G.G.H.); lcastellanosr@saludcastillayleon.es (L.C.-R.); 2Department of Health Sciences, University of Burgos, 09001 Burgos, Spain; snr1005@alu.ubu.es (S.N.-R.); ccollazo@ubu.es (C.C.); agbustillo@ubu.es (Á.G.-B.); mjgonzalez@ubu.es (J.G.-S.)

**Keywords:** lung cancer, trend, evolution, incidence

## Abstract

**Background**: This study investigates the evolution of lung cancer in the population of Burgos over more than a decade, focusing on key variables such as age, gender, histology, and stage of diagnosis. The aim is to understand how incidence rates and patterns have changed over time, especially in terms of early diagnosis. **Methods**: Retrospective data were collected from the Burgos University Hospital using histological or clinicoradiological methods. This data collection approach enabled a comprehensive examination of lung cancer trends in the province. **Results**: The results reveal an overall decrease in lung cancer incidence rates in men, offset by a steady increase in women. Histological analysis highlights a significant increase in adenocarcinoma, accounting for 43% of cases in the last year studied. Despite diagnostic advances, almost half of the diagnoses were made at stage IV, with no statistically significant change from previous years, highlighting persistent challenges in early diagnosis. **Conclusions:** The findings will not only inform resource management and prevention but could also have a significant impact on improved screening strategies and future lung cancer research.

## 1. Introduction

Lung cancer is one of the most significant global health challenges [1,2,3,4,5,6]. This is reflected in the Global Cancer Observatory report, GLOBOCAN 2020, which shows that lung cancer contributes to 11.4% of all cancer cases and 18% of cancer deaths worldwide [7]. With figures of 2.21 million incident cases and 1.8 million deaths in 2020 [8], lung cancer is consolidated as the leading cause of cancer mortality worldwide. In the Spanish context, lung cancer is no stranger to this scenario [9]. According to the latest press release from the National Institute of Statistics, 22,727 lung cancer deaths were registered in Spain in 2022, representing 19.7% of all cancer deaths [10]. Similar to global trends, lung cancer in Spain stood out as the leading cause of cancer death in men, contributing to 24.7% of cancer deaths, and the second leading cause of death in women, after breast cancer, accounting for 12.7% [11]. In Spain, statistics are available detailing the cases of lung cancer per year. These statistics also illustrate the evolution of cancer incidence rates in both absolute numbers and differences by gender. In the year 2000, there were 17,363 cases of lung cancer reported in Spain, with 15,477 cases among males and 1889 among females. This number has steadily increased over the years, reaching 22,187 cases in 2016, with 17,624 cases among males and 4563 among females [10].

In terms of gender differences in incidence rates, it is worth noting that the gender gap in lung cancer mortality has narrowed significantly in recent years [12]. While the incidence rate is decreasing in men, it has increased significantly in women [5,13], evidencing a change in the most important risk factor for lung cancer, smoking [14], as women started smoking tobacco in larger quantities later, and also took longer to quit [2]. Changes in smoking patterns date back to the 1970s, with increasing prevalence rates in women peaking in the 1990s [15,16]. This phenomenon contributes significantly to the projected increase in the incidence rate and peak rate of lung cancer among women projected for 2025, as prevalence among men and women have become increasingly similar in the younger generation worldwide [16].

Similarly, histology trends have also changed, in part, due to changing smoking patterns. Histological types closely associated with smoking, such as squamous cell carcinoma and small-cell carcinoma, have decreased in relevance globally [17]. However, in Spain, a continuation of the squamous type is observed in men [15]. On the other hand, adenocarcinoma, with a minor association with smoking [18], has emerged as the dominant histological type in both sexes [17], accounting for practically all the increase in incidence rates in women [18]. These variations in histological distribution are associated with changes in cigarette design and composition [15].

On the other hand, the evolution of lung cancer has also been dynamic in terms of diagnosis and age at disease onset. Lung cancer, in addition to its association with smoking, has a significant correlation with age, with the median age of diagnosis being around 70 years [19]. It is noted that the incidence is clearly decreasing in men under 70 years of age. In contrast, the decrease in women is only observed in those younger than 60 years [15]. With an overall 5-year survival rate of 10–20% [9], early detection becomes crucial to improve treatment options and survival outcomes [19]. However, despite advances in understanding risk factors [1], and screening recommendations for high-risk populations [20] (adults aged 50–70 years with a smoking history of more than 20 packs per year) [21], significant gaps persist, with diagnosis at later stages of disease being the norm [9,22,23].

The aim of this study is to examine the evolution of lung cancer in the population of Burgos in recent years, focusing on variables such as age, gender, histology and stage of diagnosis. Through a retrospective analysis, the aim is to provide valuable information that will contribute to the efficient management of resources, more specific prevention strategies and guidance for future research in the field of oncology.

## 2. Materials and Methods

### 2.1. Design of This Study

This longitudinal study was conducted in the city of Burgos over a period of 16 years (2000–2016), with the aim of prospectively analyzing the incidence, characteristics and trends of lung cancer in the city. Data were collected at four different points every four years (2000, 2004, 2008, 2012, and 2016), with the aim of prospectively analyzing the incidence, characteristics, and trends of lung cancer in the city of Burgos. This longitudinal approach allowed for a comprehensive examination of lung cancer dynamics over time. The population of interest included all patients diagnosed with lung cancer by histological or clinicoradiological methods during this period. Consequently, our study is classified as prospective, as data were collected over time, allowing for a continuous analysis of the incidence and characteristics of lung cancer in the studied population.

### 2.2. Participants

The study population included all patients diagnosed with lung cancer by the Pneumology unit of the Burgos University Hospital, in Spain, during the years 2000, 2004, 2008, 2012 and 2016. This four-year interval sampling approach allowed us to capture variability in lung cancer incidence rates over time and to examine possible trends based on variables such as gender, histology, diagnostic stage and mean age at diagnosis.

The study inclusion criteria encompassed patients diagnosed with lung cancer through rigorous histological or clinicoradiological methods. Specifically, patients were required to have definitive histopathological confirmation of lung cancer or a clinical diagnosis supported by radiological findings consistent with lung malignancy. This approach ensured the inclusion of accurately diagnosed cases, thus contributing to a comprehensive representation of lung cancer cases in the study population.

The voluntary participation of individuals, providing information on the collection of personal data, is noteworthy. Ethical and privacy principles were respected, and participants gave informed consent for inclusion in this study.

This methodological approach to the selection of participants ensures the representativeness of the sample, as well as the validity of the results obtained in relation to the evolution of lung cancer in the town of Burgos during the study period.

### 2.3. Procedure

Data collection was carried out systematically and uniformly using a template specifically designed for this study. In this way, relevant patient information was accurately captured. The template included several categories covering key aspects for the analysis of this study: gender, age (was categorized into specific ranges to allow for a more detailed analysis. The age groups were defined following the methodological statements of Ingram C. and Scheaffer R. [24] on consistent estimation of age replacement intervals <30 years, 30–39 years, 70–49 years, 50–59 years, 60–69 years, 70–79 years, 80–89 years or ≥90 years), histology of lung cancer (epidermoid, adenocarcinoma, microcytic, large cell, or other) and stage at diagnosis (the stage of each case at diagnosis was recorded using the standard classification into stage I, II, III or IV).

The structure of the template and its use in a standardized manner allowed for homogeneous data collection, guaranteeing the consistency of the information collected. The inclusion of these specific categories was aligned with the research objectives, facilitating the detailed analysis of key variables related to the evolution of lung cancer in the population of Burgos.

The research plan received approval from the IR Approval Clinical Research Ethics Committee of the Burgos and Soria Area. Data collection was conducted in collaborating centers by assigned personnel. Prior to sharing with the research team, all data were anonymized, ensuring anonymity and aggregation throughout the subsequent analysis.

### 2.4. Statistical Analysis

Statistical analysis was performed using SPSS version 25 statistical software (IBM-Inc., Chicago, IL, USA). Descriptive methods were used, including the presentation of a table with the main sociodemographic and clinical data, as well as a series of graphs showing the evolution of the different variables over time.

For inferential data, chi-square tests were performed to assess possible associations between the different categorical variables. The year of diagnosis was associated with patient age, gender, cancer typology, and cancer stage. The association between gender and cancer typology was also analyzed. To determine significant differences between expected frequencies and observed frequencies, absolute values greater than 1.96 or −1.96 in the corrected residuals were considered.

Data presentation was in terms of number of cases and percentage of the total for categorical variables, while continuous variables were expressed as the mean ± standard deviation of the mean.

## 3. Results

### 3.1. Sociodemographic and Clinical Characteristics of the Sample

Table 1 and Figure 1 describe the main sociodemographic and clinical characteristics of the patients diagnosed with lung cancer, as well as their percentage evolution during the years analyzed. To visualize whether the trends in the graphs were significant, the differences between the years 2000 and 2016 were analyzed. The *p*-value was included in the footer of Figure 1.

The total study sample consisted of 658 patients, of whom 560 (85.11%) were male and 98 (14.89%) were female. The mean age was 66.55 ± 11.58 years.

As can be seen in Table 1 and in the first graph in Figure 1, men represented a much higher percentage than women in all the years studied. However, it can be seen that the percentage of men is decreasing and the percentage of women is progressively increasing as the years go by.

With regard to age, hardly any cases of lung cancer were diagnosed in the population under 40 and over 89 years of age. In contrast, most of the cases diagnosed were concentrated in patients aged between 50 and 79 years.

The evolution of the type of cancer has also changed over the years. Some types, such as epidermoid and small-cell lung cancer, have maintained similar percentages between the first and last year analyzed. However, while large-cell lung cancer has decreased, adenocarcinoma of the lung has increased considerably, accounting for 42.97% of all diagnosed cases.

In almost half (49.09%) of the total cases diagnosed during all years analyzed, the stage of the disease was already stage IV, 29.18% in stage III, 6.37% in stage II, and 9.27% still in stage I. In the remaining cases (6.08%), the stage of the disease could not be determined. In the remaining cases (6.08%) the stage of the disease could not be determined.

### 3.2. Age Evolution

Table 2 below shows the inferential results obtained after associating age groups with years of lung cancer diagnosis.

As can be seen, the *p*-value is greater than 0.05, which means that there is no statistically significant difference between the expected frequencies and the observed frequencies as a function of the age of the patients diagnosed with lung cancer.

### 3.3. Gender Evolution

Table 3 below shows the inferential results obtained after associating patient gender with years of diagnosis.

As can be seen in Table 3, there was a statistically significant association (*p* = 0.020) between the variables. After analyzing the corrected residuals, significant in the years 2000 and 2016, the male count was higher and the female count was lower than expected in 2000, and the opposite in 2016, i.e., the male count was lower and the female count was higher than expected. These results reveal a decrease in lung cancer incidence rates in men, offset by an increase in women.

### 3.4. Type of Cancer Evolution

Table 4 below shows the inferential results obtained after associating lung cancer typology with years of diagnosis.

As can be seen in Table 4, there was a statistically highly significant association (*p* = 0.0001) between the variables. After analyzing the corrected residuals, there were hardly any differences between the expected count and the count obtained in the diagnoses of epidermoid and microcytic cancer, which were maintained throughout the years analyzed. In contrast, adenocarcinoma counts were lower than expected in the first years, and higher than expected in the last year, 2016. This means that the incidence of adenocarcinomas has been increasing over the years. On the other hand, the count of large-cell lung cancer was higher than expected in 2008, and much lower than expected in 2016, with not a single case diagnosed. This means that the incidence of large-cell cancer has declined considerably until the last year studied.

### 3.5. Cancer Stage

Table 5 below shows the inferential results obtained after associating cancer stage or phase with years of diagnosis.

In the table above, the aim is to analyze how the time of diagnosis, which has an impact on the stage of diagnosis, has evolved over the years. As can be seen in Table 5, there was a statistically significant association (*p* = 0.004) between the variables. After analyzing the corrected residuals, there were fewer than expected cases diagnosed in stage I during 2004, and more than expected during 2008. There were also more cases than expected in stage IV during 2004, in stage II during 2016, and in stage III during 2020.

### 3.6. Association between Gender and Type of Cancer

Table 6 below shows the inferential results obtained after associating gender with the type of cancer diagnosed.

As can be seen in Table 6, there was a statistically highly significant association (*p* = 0.0001) between the two variables. After analyzing the corrected residuals, the re-count of men diagnosed with epidermoid cancer was higher than expected, and the re-count of those diagnosed with adenocarcinoma was lower than expected. Conversely, the number of women diagnosed with squamous cell carcinoma was lower than expected, and the number of those diagnosed with adenocarcinoma was higher than expected. These results indicate that there was an association between gender and lung cancer type. Men were more likely to have epidermoid cancer and less likely to have adenocarcinoma than women.

## 4. Discussion

Lung cancer is one of the most significant challenges globally, as it is the leading cause of cancer death in men and the second leading cause in women [11]. The main objective of this study was to study the evolution of lung cancer in the population of Burgos (Spain) from 2000 to 2016, in order to analyze the main characteristics of the population, its incidence and other related aspects.

Last year’s data showed a considerable increase in women diagnosed with lung cancer, along with a percentage decrease in men diagnosed. Similar results to those obtained in other studies have also shown a recent significant increase in women [5,13]. The most likely cause is the increase in the main risk factor for lung cancer in women, smoking, as women have been modifying their habits and progressively increasing their tobacco consumption [2,14], leading to this increase in lung cancer incidence rates.

As global trends show, due to changes in smoking patterns, adenocarcinoma, despite having a minor association with smoking, has emerged as the dominant histological type in both sexes, especially in women [17,18], which may be directly related to the evolution and changes in cigarette design and composition [15]. The results of this study also showed a clear evolution in terms of disease histology. Large-cell lung cancer even disappeared in 2016, while adenocarcinoma progressively increased to represent almost 43% of all cases diagnosed in 2016. Moreover, it was also women who were more likely to suffer from adenocarcinoma.

Other studies show that smoking correlates with age, with the mean age of diagnosis being 70 years [19]. In this study, the mean age was lower in all years analyzed, and hardly any cases were diagnosed in the population younger than 40 years and older than 89 years. The majority of diagnosed cases were concentrated in patients aged 50–79 years, where screening for early detection becomes essential to increase treatment options and survival options [19]. However, very significant global health challenges remain, as it is common to diagnose the disease at advanced stages [22,23]. Nearly 80% of the total cases diagnosed in this study were stage III or IV, a condition that significantly limits survival options, increasing mortality.

The main limitations of this longitudinal study were the small number of patients diagnosed, as a consequence of being a single-center study, and the lack of more up-to-date data on the incidence of the disease.

As future lines of research, it is proposed to continue updating the data, and to increase the study population sample, including patients from other areas, in order to generalize the results obtained to the entire national and even global population. Furthermore, future research could explore the incorporation of new methods of molecular diagnostics and analysis, such as next-generation sequencing (NGS) and the analysis of circulating cells and DNA, to enhance our understanding of lung cancer evolution and improve diagnostic and treatment strategies.

## 5. Conclusions

The comprehensive analysis conducted in this study provides illuminating insights into the evolving landscape of lung cancer incidence rates, revealing nuanced trends that warrant careful consideration. Notably, while there is a discernible downward trajectory observed in lung cancer rates among men, there exists a disconcerting upward trend among women, signaling a shift in the demographic distribution of this disease. Within the realm of histological analysis, the prominence of adenocarcinoma as a predominant subtype, comprising a significant 43% of cases in the latest year under examination, underscores the shifting epidemiological profile of lung cancer.

Moreover, despite considerable advancements in diagnostic modalities, the persistent challenge of a substantial proportion of diagnoses occurring at stage IV remains troubling, highlighting the enduring obstacles in achieving timely detection and intervention. This underscores the urgent need for continued efforts aimed at improving early detection strategies and enhancing accessibility to screening programs.

In light of these findings, it becomes increasingly evident that a multifaceted approach is imperative to effectively address the complexities of lung cancer. This entails not only further research to elucidate the underlying factors driving these shifting incidence patterns but also the implementation of targeted interventions aimed at bolstering prevention efforts and optimizing resource allocation.

By fostering a deeper understanding of these evolving trends and their underlying determinants, we can better inform policy decisions, allocate resources more efficiently, and ultimately mitigate the burden of lung cancer on individuals and communities. Through concerted research endeavors and collaborative initiatives, we can pave the way for improved outcomes and a brighter future in the fight against lung cancer.

## Figures and Tables

**Figure 1 jcm-13-01601-f001:**
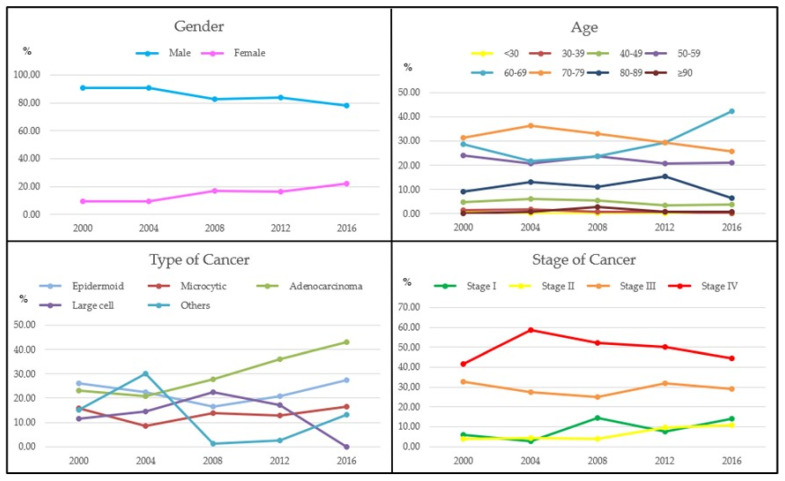
Evolution of gender, age, type, and stage of cancer. Analysis by gender between 2000 and 2016: *p*-value = 0.005. Analysis by age between 2000 and 2016: *p*-value = 0.249. Analysis by type of cancer between 2000 and 2016: *p*-value = 0.0001. Analysis by stage of cancer between 2000 and 2016: *p*-value = 0.052.

**Table 1 jcm-13-01601-t001:** Sociodemographic and clinical characteristics of the sample.

	2000	2004	2008	2012	2016
Diagnosed patients, *n*	146	116	152	116	128
Gender					
Male, *n* (%)	132 (90.4)	105 (90.5)	126 (82.9)	97 (83.6)	100 (78.1)
Female, *n* (%)	14 (9.6)	11 (9.5)	26 (17.1)	19 (16.4)	28 (21.9)
Age, Mean ± SD	65.14 ± 11.58	67.16 ± 11.67	67.03 ± 11.80	68.03 ± 11.07	65.69 ± 9.18
<30 years, *n* (%)	1 (0.7)	0 (0.0)	0 (0.0)	0 (0.0)	0 (0.0)
30–39 years, *n* (%)	2 (1.4)	2 (1.7)	1 (0.7)	1 (0.9)	0 (0.0)
40–49 years, *n* (%)	7 (4.8)	7 (6.0)	8 (5.3)	4 (3.4)	5 (3.9)
50–59 years, *n* (%)	35 (24.0)	24 (20.7)	36 (23.7)	24 (20.7)	27 (21.1)
60–69 years, *n* (%)	42 (28.8)	25 (21.6)	36 (23.7)	34 (29.3)	54 (42.2)
70–79 years, *n* (%)	46 (31.5)	42 (36.2)	50 (32.9)	34 (29.3)	33 (25.8)
80–89 years, *n* (%)	13 (8.9)	15 (12.9)	17 (11.2)	18 (15.5)	8 (6.3)
≥90 years, *n* (%)	0 (0.0)	1 (0.9)	4 (2.6)	1 (0.9)	1 (0.8)
Type of cancer					
Epidermoid, *n* (%)	38 (26.0)	26 (22.4)	25 (16.4)	24 (20.7)	35 (27.3)
Microcytic, *n* (%)	23 (15.8)	10 (8.6)	21 (13.8)	15 (12.9)	21 (16.4)
Adenocarcinoma, *n* (%)	34 (23.3)	24 (20.7)	42 (27.6)	42 (36.2)	55 (43.0)
Large cell, *n* (%)	17 (11.6)	17 (14.7)	34 (22.4)	20 (17.2)	0 (0.0)
Others, *n* (%)	22 (15.1)	35 (30.2)	2 (1.3)	3 (2.6)	17 (13.3)
Stage					
I, *n* (%)	9 (6.2)	3 (2.6)	22 (14.5)	9 (7.8)	18 (14.1)
II, *n* (%)	6 (4.1)	5 (4.3)	6 (3.9)	11 (9.5)	14 (10.9)
III, *n* (%)	48 (32.9)	32 (27.6)	38 (25.0)	37 (31.9)	37 (28.9)
IV, *n* (%)	61 (41.8)	68 (58.6)	79 (52.0)	58 (50.0)	57 (44.5)

SD = Standard Deviation.

**Table 2 jcm-13-01601-t002:** Chi-square test to determine the association between age and year of diagnosis.

		2000	2004	2008	2012	2016
<30	Count	1	0	0	0	0
	Expected count	0.2	0.2	0.2	0.2	0.2
	Corrected residual	1.9	−0.5	−0.5	−0.5	−0.5
30–39	Count	2	2	1	1	0
	Expected count	1.3	1.1	1.4	1.1	1.2
	Corrected residual	0.7	1.0	−0.4	−0.1	−1.2
40–49	Count	7	7	8	4	5
	Expected count	6.9	5.5	7.2	5.5	6.0
	Corrected residual	0.1	0.7	0.4	−0.7	−0.5
50–59	Count	35	24	36	24	27
	Expected count	32.4	25.7	33.7	25.7	28.4
	Corrected residual	0.6	−0.4	0.5	−0.4	−0.3
60–69	Count	42	25	36	34	54
	Expected count	42.4	33.7	44.1	33.7	37.2
	Corrected residual	−0.1	−2.0	−1.7	0.1	3.7
70–79	Count	46	42	50	34	33
	Expected count	45.5	36.1	47.4	36.1	39.9
	Corrected residual	0.1	1.3	0.5	−0.5	−1.5
80–89	Count	13	15	17	18	8
	Expected count	15.8	12.5	16.4	12.5	13.8
	Corrected residual	−0.8	0.8	0.2	1.8	−1.8
≥90	Count	0	1	4	1	1
	Expected count	1.6	1.2	1.6	1.2	1.4
	Corrected residual	−1.4	−0.2	2.1	−0.2	−0.3

X^2^ (658) = 32.609, *p* = 0.250.

**Table 3 jcm-13-01601-t003:** Chi-square test to determine the association between gender and year of diagnosis.

		2000	2004	2008	2012	2016
Male	Count	132	105	126	97	100
	Expected count	124.3	98.7	129.4	98.7	108.9
	Corrected residual	2.0	1.8	−0.9	0.5	−2.5
Female	Count	14	11	26	19	28
	Expected count	21.7	17.3	22.6	17.3	19.1
	Corrected residual	−2.0	−1.8	0.9	0.5	2.5

X^2^ (658) = 11.631, *p* = 0.020.

**Table 4 jcm-13-01601-t004:** Chi-square test to determine the association between type of lung cancer and year of diagnosis.

		2000	2004	2008	2012	2016
Epidermoid	Count	38	26	25	24	35
	Expected count	32.9	27.5	30.5	25.6	31.5
	Corrected residual	1.2	−0.4	−1.3	−0.4	0.8
Microcytic	Count	23	10	21	15	21
	Expected count	20.0	16.7	18.5	15.5	19.1
	Corrected residual	0.8	−2.0	0.7	−0.2	0.5
Adenocarcinoma	Count	34	24	42	42	55
	Expected count	43.9	36.7	40.6	31.0	41.9
	Corrected residual	−2.1	−2.8	0.3	1.8	2.8
Large cell	Count	17	17	34	20	0
	Expected count	19.6	16.4	18.1	15.2	18.7
	Corrected residual	−0.7	0.2	4.5	1.5	−5.3
Others	Count	22	35	2	3	17
	Expected count	17.6	14.7	16.3	13.6	16.8
	Corrected residual	1.3	6.3	−4.3	−3.4	0.1

X^2^ (602) = 103.097, *p* = 0.0001.

**Table 5 jcm-13-01601-t005:** Chi-square test to determine the association between stage of diagnosis and year of diagnosis.

		2000	2004	2008	2012	2016
Stage I	Count	9	3	22	9	18
	Expected count	12.2	10.7	14.3	11.4	12.4
	Corrected residual	−1.1	−2.7	2.4	−0.8	1.9
Stage II	Count	6	5	6	11	14
	Expected count	8.4	7.3	9.9	7.8	8.6
	Corrected residual	−1.0	−1.0	−1.5	1.3	2.2
Stage III	Count	48	32	38	37	37
	Expected count	38.5	33.6	45	35.7	39.1
	Corrected residual	2.1	−0.4	−1.4	0.3	−0.5
Stage IV	Count	61	68	79	58	57
	Expected count	64.8	56.4	75.8	60.1	65.9
	Corrected residual	−0.8	2.5	0.6	−0.4	−1.8

X^2^ (618) = 28.826, *p* = 0.004.

**Table 6 jcm-13-01601-t006:** Chi-square test to determine the association between gender and cancer type.

		Epidermoid	Microcytic	Adenocarcinoma	Large Cell	Others
Male	Count	142	77	146	76	69
	Expected count	125.4	76.2	166.9	74.6	66.9
	Corrected residual	4.4	0.2	−5.0	0.5	0.7
Female	Count	6	13	51	12	10
	Expected count	22.6	13.8	30.1	13.4	12.1
	Corrected residual	−4.4	−0.2	5.0	−0.5	−0.7

X^2^ (602) = 32.181, *p* = 0.0001.

## Data Availability

All the study data can be found in the following link: https://docs.google.com/spreadsheets/d/16BmSIdBc9e3kPCHpJ9TlZqK6qz8dqszP/edit#gid=1596017488.

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
