# Peer review of "Retrospective Study of Lung Cancer: Evolution in 16 Years in the Burgos Region"

_jcm, 2024, doi:10.3390/jcm13061601_

Round 1

Reviewer 1 Report

Comments and Suggestions for Authors

This is an interesting article. From my perspective there are some questions:

1.      “This longitudinal study was conducted in the city of Burgos over a period of 16 years

(2000-2016), with the aim of prospectively analysing the incidence, characteristics and

trends of lung cancer in the city. The population of interest included all patients diag-

nosed with lung cancer by histological or clinic or adiological methods during this period”

Does it mean that the study started in 2000 and ended in 2016, or do authors only mean that they use data from 2000-2016?   This is relevant as this should reflect terms “prospective” and “retrospective”. 

2.      In 2.4 “For inferential data, chi-square tests were performed to assess possible associations

between the different categorical variables”. Here authors should describe/mention which associations they analyzed.

3.      Figure 1 shows trends but does not display if these trends were significant. This is however very important in trend analyse, especially based on relatively small samples per unit.  At least the difference 2016 vs. 2000 should be tested.

4.      Do authors have statistics from whole Spain (in Germany , federal statistical office published such data early) to compare Burgos with Spain in total to see if this region differs?

Reviewer 2 Report

Comments and Suggestions for Authors

Brief Summary: The manuscript titled "Retrospective Study of Lung Cancer: Evolution in 16 years in the Burgos Region" aims to investigate the trends in lung cancer incidence, histology, and staging over 16 years in the Burgos region. The study makes significant contributions by highlighting changes in lung cancer patterns and emphasizing the need for further research, resource management, and prevention strategies. The strengths of the paper lie in its clear presentation of data, relevant statistical analysis, and consistent interpretation of results.

General Comments: The article effectively addresses the evolution of lung cancer in a specific region over a significant timeframe. However, there are areas of weakness that need to be addressed. The hypothesis testing could be strengthened by providing more detailed information on the selection criteria for participants and potential biases in the study population. Methodological inaccuracies may arise from the need for more information on potential confounding variables or factors that could influence the observed trends. Including controls or comparative data from other regions could enhance the robustness of the study.

Review Comments: The manuscript comprehensively covers the topic of lung cancer evolution, focusing on key variables such as age, gender, histology, and stage of diagnosis. The relevance of the review topic is evident in the context of understanding changing patterns of lung cancer incidence and the challenges in early diagnosis. The study effectively identifies a gap in knowledge regarding the need for updated data on disease incidence and the generalizability of results to a broader population. The references cited are mostly recent and relevant, contributing to the credibility of the study findings.

Specific Comments:

·      Line 72: Clarify the specific criteria used for participant selection in the study.

·      Line 72: Design, instead of Desing

·      Table 5: Provide more detailed descriptions of the statistical tests used for the associations presented in the table.

·      Line 261: Consider adding a statement on data availability for transparency and reproducibility purposes.

·      Figure 1: Ensure that the figure legend clearly explains the variables represented in the graph for better interpretation by readers.

Overall, the manuscript presents valuable insights into the evolution of lung cancer in the Burgos region. Addressing the methodological weaknesses and providing additional clarity on participant selection criteria and statistical analyses will strengthen the scientific rigor of the study. 

Comments on the Quality of English Language

A moderate level of editing is required. Advise authors to use an English editing tool such as Grammarly to proofread.

Reviewer 3 Report

Comments and Suggestions for Authors

please better explain in the methods the type of study.

Please explain the reason why the patients were categorized according to the age  as follows: <30 years, 30-39 years, 70- 49 years, 50- 99 59 years, 60-69 years, 70-79 years, 80-89 years or ≥90 years),

It is noteworthy to point out about the epidemiological modification of the histotype of lung cancer associated with variations of tobacco consumption

It would be interesting  to discuss about new methods of molecular diagnostics and analysis including NGS and analysis of cells and circulating DNA

J Thorac Dis. 2018 Jul;10(7):E570-E576.

Round 2

Reviewer 2 Report

Comments and Suggestions for Authors

Accept in present form.